# Does social media information credibility influence social commerce purchase intention of skincare products? Evidence from Facebook

Prarthana Ranjith[1◉], Sumudu Nisansala[1◉], Nimesha Jayasingha[1◉],
Kavindya Weerasekara[1◉], Krishantha Wisenthige[2‡*], Nirmani Dayapathirana[2‡]

1 Department of Information Management, Sri Lanka Institute of Information Technology, Malabe, Sri Lanka, 2 Department of Business Management, Sri Lanka Institute of Information Technology, Malabe, Sri Lanka

◉ These authors contributed equally to this work.
‡ KW and ND also contributed equally to this work.
* krishantha.w@sliit.lk

## Abstract

Social commerce is transforming consumer purchasing behaviours by blending social media interactivity with e-commerce functionalities, and most purchases today are evidently facilitated through social media platforms with ease. Recognising the importance of credibility in skin-related purchases, this study aims to examine how social media information credibility factors, specifically source credibility and electronic word of mouth (e WOM) credibility, influence consumers' purchase intentions for skincare products on Facebook, considering the mediating roles of trust in online communities and perceived privacy risk. Primary data were collected through a structured survey from 384 skincare purchasers who made their purchases via Facebook, and the model was tested using structural equation modelling (SEM). Further, the results reveal that source credibility, e WOM credibility, and trust in online communities positively influence social commerce purchase intention (SCPI), while perceived risk has a negative effect. Trust in online communities also reduces perceived risk and mediates the relationship between information credibility and purchase intention. Hence, these findings highlight the pivotal roles of trust and risk perceptions in shaping online consumer behaviour in the social commerce space, especially within the skincare market. The study emphasises the need for businesses to leverage credible information sources and build trustworthy online communities to enhance consumer confidence and engagement. Moreover, it contributes to the growing literature on social commerce by offering insights from an emerging market context, Sri Lanka, and suggests future research into broader dimensions of credibility and cultural comparisons to deepen the understanding of social commerce.

**Data availability statement:** All relevant data are within the manuscript and its Supporting information files.

**Funding:** The author(s) received no specific funding for this work.

**Competing interests:** The authors have declared that no competing interests exist.

## Introduction

"Credibility of information shared on social media is the foundation of consumer confidence in social commerce, shaping purchase decisions and fostering long-term business relationships" [1,2]. By the end of 2024, over 5.2 billion people worldwide used social media, representing 63.8% of the global population [3]. This surge in usage highlights consumers' increasing dependence on digital platforms for daily activities [4–6]. Currently, social media has shifted from a communication and information-sharing platform to a powerful tool for marketing and sales [7–10]. Moreover, social media has become a significant avenue for social commerce, enabling consumers to explore products and interact with online businesses through reviews, recommendations, and user-generated content [6,11,12]. The relationship between social media platforms and their influence on consumer purchasing intention has emerged as a notable global trend, shaping purchase decisions across numerous situations [6,10,13,14].

There are many social media platforms, but Facebook is one of the most popular and important for social commerce. With 3.2 billion active users worldwide, Facebook also remains the leading platform in Sri Lanka, with approximately 7.45 million active users as of 2024, accounting for over 34% of the national population [15]. It allows businesses to reach a large audience and directly interact with customers [16,17]. Its wide reach, easy-to-use features, and marketing tools make it a major player in social commerce. The large number of users and its ongoing popularity demonstrate the importance of Facebook in helping businesses connect with customers and increase sales in the digital world.

Like other industries, the skincare market is experiencing rapid growth driven by the increasing efficacy of skincare products and treatments. Sri Lanka's skincare industry has shown consistent growth in recent years, fueled by rising health consciousness, urbanisation, and increased social media usage among younger demographics [18]. Platforms such as Facebook and Instagram have become vital channels for skincare brands to engage consumers through influencer campaigns, user-generated content, and direct-to-consumer promotions [18,19]. However, the industry faces credibility challenges due to limited regulatory oversight and the proliferation of unverified product claims online [20]. These dynamics make the skincare sector in Sri Lanka a critical context for examining trust and information credibility in social commerce [18,20,21].

User-generated content (UGC), such as reviews, ratings, testimonials, and social media comments, plays a central role in shaping consumer perceptions of credibility and trust in social commerce environments [22]. Consumers often question the trustworthiness of user-generated content, such as reviews, ratings, and recommendations. Many are concerned that information on social media could be biased, exaggerated, or even false, leading to doubts about its reliability [19,22]. Research shows that 70% of online consumers worry about the credibility of information they find on social media platforms [23]. Prior research confirms that UGC significantly influences consumers' attitudes, risk perceptions, and purchase intentions, often more than firm-generated content [24,25]. User-generated content serves as a form of electronic

word-of-mouth (e WOM), where consumers perceive peer opinions as more authentic and trustworthy [26]. Despite extensive literature on user-generated content, its impact within credibility-sensitive industries like skincare and in understudied digital markets such as Sri Lanka, remains underexplored.

Previous research on online shopping behaviour has predominantly focused on general e-commerce or demographic factors such as age, gender, education, income, and geographic location [27,28]. For instance, a study in Turkey found that younger consumers with higher education and urban residence were more inclined to engage in online purchases [29]. Other studies have explored the effects of either source credibility or electronic word-of-mouth (e WOM) credibility in isolation on consumer trust or purchase decisions [24,30]. However, there is limited research that integrates these credibility dimensions within a unified framework, particularly alongside trust and perceived risk, to explain purchase intention in social commerce.

While social commerce literature has advanced in exploring individual factors like trust, engagement, and platform usability, most research has focused on general products like electronics, fashion, and services [31]. These studies are also largely conducted in developed economies, limiting their applicability to emerging markets. Little attention has been given to how credibility-related factors interact in credibility-sensitive sectors like skincare, where purchase decisions are strongly influenced by trust due to personal and health-related implications.

Although skincare brands increasingly use social media to influence purchase behaviour. Existing studies rarely assess how consumers evaluate the credibility of this content when making purchase decisions, particularly for products that require a high level of consumer trust [19]. In addition, research in developing economies such as Sri Lanka, where social media usage is rising but trust in digital content remains uncertain, is still limited.

This study addresses these gaps by proposing an integrated model that examines the influence of source credibility and e WOM credibility on purchase intention, mediated by trust in online communities and perceived risk, specifically within the skincare industry in Sri Lanka, while also incorporating demographic variables as contextual factors. Therefore, it contributes both theoretical insight and practical guidance for businesses operating in credibility-sensitive markets within emerging economies.

The motivation of this study stems from the growing reliance on social media platforms for purchase decisions and the critical need for credible information, particularly for sensitive products like skincare. Its significance lies in helping businesses in developing economies, such as Sri Lanka, build effective digital strategies to enhance consumer trust and drive social commerce success.

Accordingly, this study aims to investigate how social media information credibility factors influence social commerce purchase intention (SCPI) in Sri Lanka. The main objective is supported by specific sub-research objectives. Firstly, the study aims to evaluate the impact of source credibility on perceived risk. Secondly, it seeks to assess the impact of perceived risk on social commerce purchase intention. Thirdly, the research analyses the impact of e WOM credibility on trust in online communities. Fourthly, it examined the impact of trust in online communities on social commerce purchase intention. After that, the study evaluates the impact of trust in online communities on perceived risk. Finally, the study evaluates that the mediating impacts of perceived risk mediate the relationship between source credibility and social commerce purchase intention, while trust in the online community mediates the impact of e WOM credibility on both perceived risk and purchase intention, with a sequential mediation effect where trust influences perceived risk, ultimately affecting purchase behaviour as a mediating effect of social commerce context. These objectives collectively provide a comprehensive framework for understanding the role of information credibility in shaping consumer behaviour within the Sri Lankan social commerce context.

Moreover, the novelty of this study lies in its integrated exploration of multiple social media information credibility, specifically source credibility and e WOM credibility and their effects on social commerce purchase intention through the mediating roles of perceived risk and trust in online communities.

Despite the expanding literature on social commerce, most studies examine source credibility, e WOM, trust, and perceived risk as discrete constructs, lacking a unified framework that captures their combined impact on consumer purchase

intention, particularly in high-involvement categories like skincare, where decisions are highly trust-dependent [18]. Additionally, the trust risk relationship remains theoretically ambiguous, with prior research reporting contradictory mediating roles and causal directions [32,33]. It limits clarity on how credibility perceptions operate in sensitive decision-making contexts.

Practically, while consumers increasingly rely on social media for product evaluation, this trust is undermined, especially in developing markets like Sri Lanka, where digital adoption coexists with content scepticism. Reports cite growing concerns over influencer credibility, fake reviews, and unverifiable claims, which reduce conversion and weaken brand loyalty in sectors like skincare [34]. Addressing these gaps is urgent; thus, theoretically, it advances credibility-based models in social commerce. Therefore, practically, it informs trust-driven strategies to improve marketing effectiveness in emerging markets.

The remaining sections are structured as follows: 'Literature Review', which covers theories and prior findings on social media credibility and purchase intention. Data and Methodology explain the data collection and SEM approach. 'Results and Discussion', analysing the empirical findings and their interpretations, and "Conclusion," provide recommendations and future research directions.

## Literature review

This section offers a reflective analysis of prior literature to establish the foundation for the present study. A systematic search was carried out to identify, evaluate, and synthesize relevant studies, ensuring both breadth and depth of coverage. The search strategy, keywords, and inclusion and exclusion criteria employed in this process are summarized in Fig 1.

### Source credibility

Source credibility refers to the positive characteristics of a communicator that enhance the value and impact of information in the reception of a message [35–37]. Similarly, the study by Jackob and Hueb [37] defines source credibility as a positive

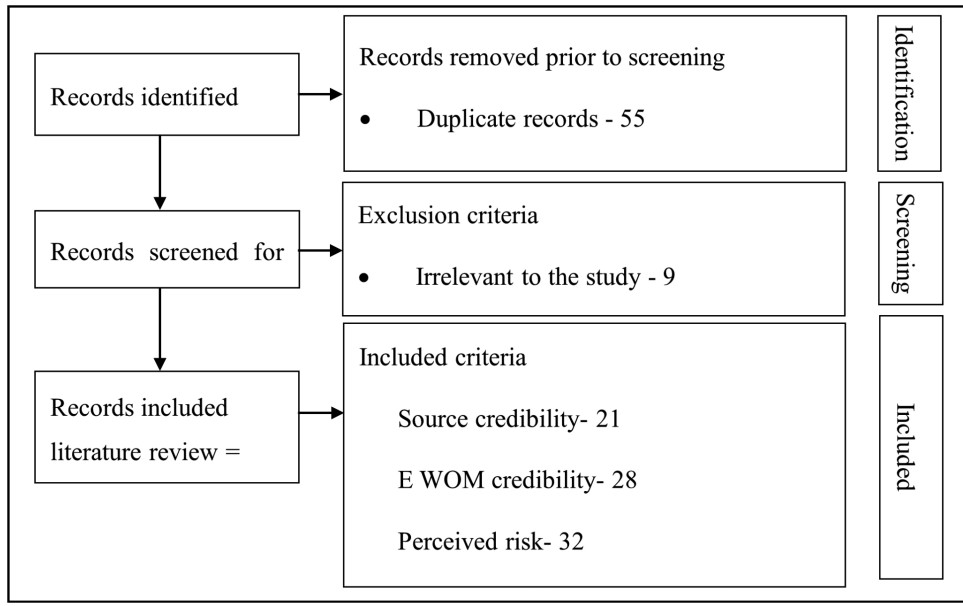

**Fig 1. Prisma flow diagram.** Source: Authors' compilation.

underlying features of a communicator that make a message more easily accepted or persuasive. It plays a crucial role in shaping social media purchase intentions. Furthermore, the study by Ismagilova, Slade [38] identified three dimensions of source credibility: source expertise, source trustworthiness, and homophily. These characteristics significantly influence consumer purchasing behaviours. However, this study specifically focuses on source trustworthiness and source expertise. Consumer attitudes toward the credibility of information sources are critical factors that drive purchase intentions [30,39–41]. Recent research reconfirms that source expertise, or the extent to which the source is perceived as knowledgeable, and source trustworthiness, or the perception of honesty and reliability, are essential for consumer perception and purchase behaviour [38,41]. Consumer attitudes regarding the credibility of online information sources can stimulate purchase intention through scepticism and uncertainty reduction [17,42].

Previous research indicates that source credibility negatively impacts individuals' perceived risk [43–45]. Additionally, studies by Balouchi, Aziz [46], have demonstrated a significant relationship between source credibility and perceived risk in the context of online purchasing. Still, the analysis is lacking in the context of skincare, especially within social commerce sites such as Facebook. This study seeks to address this concern by exploring the influence of source expertise and trust on perceived privacy risk regarding purchasing skincare products on Facebook in Sri Lanka. To address this gap, the following hypothesis is proposed:

$H_1$: Source credibility has a negative significant impact on perceived risk.

## e WOM credibility

Traditionally, word-of-mouth was defined as the oral exchange of information through interpersonal, non-commercial interactions among mutual peers [24,42,47–49]. With the advancement of social media over time, traditional word-of-mouth communication has evolved significantly. This form of non-commercial communication has been transformed into e WOM, a new mode of communication. e WOM takes various forms, including consumer opinions, recommendations, comments, and product reviews shared on social media platforms. e WOM has garnered significant attention in fields such as marketing and consumer behaviour due to its ability to influence commerce.

Early research explored various aspects of e WOM, but limited studies have specifically examined the impact of e WOM credibility on purchase intentions in a social commerce setting [6]. e WOM is characterised by recommendations, reviews, and comments in an online context [50,51]. According to Cheung and Lee [24], the study has emphasised that e WOM elements comments, reviews, and recommendations, directly impact consumers' buying incentives in social commerce. e WOM functions as a channel for customers to share marketing information, significantly influencing consumer attitudes and behaviours toward products or services. The increasing popularity of social media, driven by the advent of Web 2.0, has made social media content a vital information resource for consumers making purchase decisions on social commerce platforms [52,53].

However, to the best of researchers' knowledge and relative to studies dealing with the e -WOM credibility and trust nexus in Online Communities focused on skincare products in Facebook boards and along emerging economies in the Indian region, women from Sri Lanka. Nevertheless, most of the studies have focused on broad product categories or centred on advanced economies. These studies are worth addressing because decades of building communication information technology have significantly boosted e WOM elements, including reviews, comments, and recommendations, and positively impacted consumers' trust, which certainly strengthens their purchase intentions [6]. As such, the study focuses on how e WOM credibility impacts the level of consumer trust in skincare Facebook groups, where social interaction and social commerce greatly affect each purchase decision [6,17,54]. To address this gap, the following hypothesis is proposed:

$H_2$: e WOM credibility has a positive significant impact on trust in online communities.

## Perceived risk

In marketing, perceived risk refers to "the nature and amount of risk perceived by a consumer in contemplating a particular purchase action or behaviour" [55]. In an online context, Dowling and Staelin [51], defined perceived risk as "consumers' attitudes or perceptions of the uncertainty and adverse consequences of buying a product or service". According to Ali, Abbass [42], further explained perceived risk as "the likelihood that individuals gather information online and use it most appropriately". The buyer's psychological evaluation of potential losses associated with online purchases is central to the concept of perceived risk in social commerce, which relates to "the consumer's view of the uncertainty and negative effects of purchasing products or services".

According to Ali, Abbass [42], consumers who have not made online purchases are more likely to avoid doing so, as perceived risk significantly influences online buying behaviour. A prior study by Yan, Wu [56], examined how different aspects of perceived risk have varying effects on consumer behaviour in China from survey data. In social commerce, perceived risk impacts consumer satisfaction and their likelihood of returning for future business. Similarly, prior studies have shown that perceived risk strongly influences purchase intention, often interacting with factors like trust and intimacy in social commerce settings [57].

Perceived risk provides insights into consumer behaviour in online purchasing environments, where uncertainty in e-commerce and s-commerce often shapes buying decisions. A previous research study by Wang, Wang [58], investigated into South African online market and found that perceived risk affects online buying behaviour, with consumers lacking prior online purchase experience being less likely to engage in future transactions. Additionally, earlier research identified multiple dimensions of perceived risk in online shopping, including financial, social, time, product, delivery, after-sales, security, and privacy risks.

Past studies, as referenced in Duan, Chen [6] and Breward [59], consistently show the negative impact risk perception has on purchase intention in both e-commerce and social commerce settings. As for social commerce platforms, perceived risks, including privacy risk and delivery risk, significantly decrease user engagement and willingness to purchase items through social platforms [17]. However useful these insights are, the literature regarding skincare purchase in the context of social commerce in Sri Lanka is silent on how trust toward a platform, credibility of a source, and e WOM influence that relationship through moderation or mediation. To address that gap, this research aims to examine how perceived privacy risk affects purchase intention within Facebook skincare communities. Based on these findings, this study proposes the following hypothesis:

H$_3$: Perceived risk has a negative significant impact on social commerce purchase intention.

## Trust online community

Inspired by the concept of trust as "existing when one party has confidence in the exchange partner's reliability and integrity" [60]. Trust is considered a crucial component of effective relationship marketing, relying on commitment and confidence between parties. Trust has also been defined as "the willingness of a party to be vulnerable to the actions of another party based on the expectation that the other will perform a particular action important to the trustor, irrespective of the ability to monitor or control that other party" [61]. Similarly, the study by Pavlou [33], described trust as "the belief that the other party will behave in a socially responsible manner and fulfil the trusting party's expectations without exploiting its vulnerabilities".

Users who trust an online community are more likely to accept unsolicited advice and be assured about the items they buy [6,24]. The growing interactions of consumers with products through reviews, ratings, and comments make trust an essential mediator in navigating the information credibility and the actions taken thereafter. Trust enables consumers to feel safe even when there is no direct engagement or immediate remedy available in case of failure of transactions. Regardless of the importance of trust, it remains one of the hurdles for social commerce in developing

countries. Problems such as late delivery, fake goods, and bad after-sales service damage the trust and thus the purchase intention.

From a customer perspective, the dimensions of trust commonly include competence, benevolence, and integrity [62]. Benevolence refers to the likelihood that a company acts consistently, reliably, and honestly in fulfilling promises, while competence is the company's ability to deliver on its commitments with genuine concern for customer welfare. Trust is a significant factor influencing purchase intentions and reducing perceived risk. However, some social commerce initiatives have faced challenges related to trust, security, and privacy, resulting in consumer complaints [63]. Nonetheless, many sellers have effectively utilised social commerce to enhance their business activities [10].

Building trust in online communities is critical to alleviating consumer concerns about commercial exchanges and positively influencing purchase intentions in social commerce settings [17,42,63,64]. Online communities allow consumers to share information about products and services, helping to foster trust, which in turn impacts their purchasing decisions [11,65]. Prior findings indicate that trust in online communities positively affects social commerce purchase intentions and consumer purchasing behaviour [17,66,67].

Past Studies have also established a link between trust in online communities and customer loyalty [10,68–70]. Consumers with strong trust in online communities are more likely to be influenced to purchase products recommended within these communities. While trust has a positive effect on purchase intention, it also negatively impacts perceived risk. Additionally, trust in social commerce settings has an adverse impact on perceived risk, especially when it comes to privacy and security concerns, which are two major obstacles to online purchases [41,42,71]. By reducing perceived risk, trust directly enhances consumer purchase intention in online settings [32,72]. However, some social commerce services have successfully used community interactions and credible e WOM to build trust and reduce risk [29]. The focus of this research is trust within skin care Facebook groups in Sri Lanka, where it is argued that with increased trust, purchase intention is likely to increase while perceived risk will decline. Based on these findings, the following hypotheses are proposed:

$H_4$: Trust in online communities has a negative significant impact on perceived risk.

$H_5$: Trust in online communities has a positive significant impact on social commerce purchase intention.

## Social commerce purchase intention

According to Hajli, Sims [63], social commerce, a subset of electronic commerce, has recently emerged as a significant area of study. Social media commerce has become a central focus of research in recent years. While some studies have examined how social media platforms influence consumers' purchase intentions, others have explored the role of social interactions in shaping consumer behaviour [5,73]. For example, the prior research by Mikalef, Giannakos [2], investigated how the features of social commerce platforms impact users' purchase intentions and participation in word-of-mouth communication. While Jackob and Hueb [37], analysed the effect of social media platform usage on users' purchase intentions.

Social commerce involves not only buyers and sellers but also individuals with social connections on the platform, which influences the flow of information on social media. According to Algharabat and Rana [74], revealed that consumers' purchase intentions can be significantly enhanced through instrumental inter-user connections, which improve their perception of information and product quality. With social networking sites and social media enabling consumers to actively create content online, social commerce represents a novel and appealing development in e-commerce. Social media plays a crucial role in distinguishing social commerce from traditional e-commerce by fostering active user interaction and engagement [17,73,75].

Social commerce leverages Web 2.0 applications to enable interactions among users in an online environment, where their contributions can facilitate the acquisition of products and services [76]. While related research has been conducted

on social media commerce. According to Zhang, Law [5], noted that few studies have specifically examined how information (content) affects consumer purchase intentions. This is particularly significant on social media platforms, where customers cannot physically inspect the goods they are considering and must rely on information provided by sellers and intermediaries.

This gap emphasises the need to study the influence of information credibility on purchase intentions in the social context within content-led social commerce [30,61,77]. Gaining this understanding is important in shedding light on how consumers with asymmetrical information manage to trust decisions made in online purchases, especially those regarded as high involvement, like skincare products.

## The mediating role of trust online community and perceived risk

In examining the mediating effects within social commerce, three critical pathways were identified to explore the indirect influences of credibility factors on social commerce purchase intentions. Following the guidelines of Hayes, Montoya [78], the mediation process requires establishing relationships between the independent variables (IV), mediating variables (MV), and the dependent variable (DV). Importantly, a direct relationship between the IV and DV is not a prerequisite for significant mediation, as indirect effects can exist independently of direct effects.

The first mediation pathway involves source credibility, which significantly reduces the perceived risk associated with online transactions. This reduction in perceived risk, in turn, enhances social commerce purchase intention. Previous studies suggest that higher source credibility fosters a sense of reliability and lowers the psychological uncertainty surrounding online shopping decisions [46,79]. The second mediation pathway incorporates e WOM credibility, which positively influences trust in the online community. This heightened trust reduces perceived risk, ultimately leading to an increase in social commerce purchase intention. Trust in the online community serves as a critical mediator in this process, bridging the gap between e WOM credibility and risk perceptions in social commerce contexts [24,80]. Lastly, e WOM credibility also directly impacts trust in the online community, which further enhances social commerce purchase intention. This pathway highlights the significant role of trust as a mediator that translates the informational value of e WOM into consumer confidence and purchase behaviour [42,81]. Based on these mediation effects following hypotheses were developed:

$H_6$: Perceived risk mediates the relationship between source credibility and social commerce purchase intention.

$H_7$: Trust in the online community and perceived risk sequentially mediate the relationship between e WOM credibility and social commerce purchase intention.

$H_8$: Trust in the online community mediates the relationship between e WOM credibility and social commerce purchase intention.

Based on hypotheses developed from previous literature, key variables are outlined in Table 1.

## Theoretical foundation

The Elaboration Likelihood Model (ELM) of Petty and Cacioppo [82], has two branches that deal with the ways people are persuaded through messages. It is a dual process theory. Two primary forms of processing information are identified by ELM: the central route, in which participants painstakingly pay attention to message arguments; the peripheral route, where decisions are made using other cues such as source credibility or message style.

Users on social commerce platforms, e.g., Facebook, as well as Amazon, constantly expose themselves to Product reviews, consumer opinions, and experiences. Thus, ELM explains evaluating intentions to engage in budgetary behaviour. The deeper or central processing users engage in when the information credibility increases, convinces them during the perception stage, empowers more willing behavioural trust, and strongly decreases risk, improving intent.

**Table 1. Variables and sources.**

| Variables | Definition of the Variable | Dimension | Source |
|---|---|---|---|
| Source Credibility | A communicator's positive traits enhance how their message is received. | • Expertise<br>• Trustworthiness | [46,57,65,90,91] |
| e WOM | Any positive or negative statement about a product or company made by potential, actual, or former customers and shared online with a large audience. | • Recommendation<br>• Reviews<br>• Comments | [24,67,92,93] |
| Perceived Privacy Risk | The level of risk a consumer feels when considering a purchase. | • Financial risk<br>• Time risk<br>• Product risk<br>• After-sales risk | [43,51,72,94] |
| Trust Online Community | The willingness to rely on another party, expecting them to perform a specific important action. | • Competence<br>• Benevolence<br>• Reciprocity<br>• Reliability | [61,63,95,96] |
| Social Commerce Purchase Intention | A consumer's willingness to engage in online social commerce transactions. | • Social presence<br>• Trust<br>• Social interaction | [5,74,76,77] |

Source: Authors' compilation.

Additionally, both source credibility (the trust and expertise of the individual conveying the information) and e WOM credibility (the perceived reliability and quality of online reviews) are critical under ELM in terms of persuasion. The model further accommodates the mediation effects of trust and perceived risk, which assists in describing how persuasive information is transformed into purchase intentions.

Hypotheses Development (With Theory Support)

$H_1$: Source credibility has a negative significant impact on perceived risk.

As outlined in the ELM, a credible figure serves as a persuasive cue that increases consumer confidence and reduces uncertainty. Since consumers feel more confident when reasoning comes from a knowledgeable and trusted source, their perception of risk diminishes [82].

$H_2$: e WOM credibility has a positive significant impact on trust in online communities.

Relevant e WOM communication is helpful as it is aligned with quality expectations within the framework of ELM. Trust tends to be granted by users after evaluating and deeming these messages credible, originating from an online community.

$H_3$: Perceived risk has a negative significant impact on social commerce purchase intention.

Based on ELM principles, higher levels of perceived risk tend to lower the propensity to carry out purchase activities. Central processing declines due to overwhelming doubt or anxiety because attempts to mitigate these feelings lack effectiveness purchasing intent decreases.

$H_4$: Trust in online communities has a negative significant impact on perceived risk.

Trust is crucial for online information processing. If consumers trust an online community, their uncertainty diminishes along with the risk they perceive associated with the product and platform.

$H_5$: Trust in online communities has a positive significant impact on social commerce purchase intention.

Trust fosters belief in a community's information credibility. Such trust enhances an individual's assessment of the products/services being offered, increasing the purchasing probability.

$H_6$: Perceived risk mediates the relationship between source credibility and social commerce purchase intention.

Within the ELM framework, recognising credible sources helps resolve ambiguity, which lowers the risk associated. This lower risk mediates through transforming higher purchase intention with less source risk perceptions, turning the theory upside down.

$H_7$: Trust in the online community and perceived risk sequentially mediate the relationship between e WOM credibility and social commerce purchase intention.

When consumers regard WOM as credible, they trust that community, which diminishes their perceived risk. This supports sequential mediation consistent with ELM's central route, consisting of layered cognitive evaluations driving behaviour.

$H_8$: Trust in the online community mediates the relationship between e WOM credibility and social commerce purchase intention.

Trust in the platform is gained from credibility assigned to e WOM, and this trust significantly enhances the purchase intention. Attitudinal results due to message appraisal are emphasised by ELM, which is supportive of this singular mediation route.

## Research conceptual model

Based on the motivation and purpose of this study, the impact of factors such as source credibility and e WOM credibility was conceptualised as components of social media information credibility, while investigating social commerce purchase intentions. Furthermore, how these factors influence perceived risk and trust in online communities is explored. In that context, source credibility, e WOM credibility, trust in online communities, and perceived risk are identified as major antecedent variables of social commerce purchase intention. Source credibility represents the perceptions of the source as being expert and trustworthy, which can reduce the feeling of uncertainty and has a positive influence on purchasing. Belief in and credibility of e WOM Belief in and credibility e WOM, refers to the trustworthiness, believability, and reliability of online reviews and recommendations from consumers. Online community trust is the trust that users feel towards the other members and the information shared in the social platforms, promoting purchasing behaviour. Perceived risk includes consumers' concerns about potential negative results of trading online, which is an important aspect to reduce perceived risk to positively influence purchase intention. These variables provide an inclusive model that explains the consumer behaviour in social commerce environments on the grounds of a social exchange perspective. The mediating effects of perceived risk and trust in online communities on the relationship between social media information credibility and social commerce purchase intentions are also examined. Based on these objectives, the conceptual framework in Fig 2 was established. Source credibility and e WOM credibility are identified as the independent variables in this study, social commerce purchase intention is designated as the dependent variable, and perceived risk and trust in online communities are considered the mediating variables.

## Method

### Research design

In this case, the social media information credibility as an indicator of social commerce purchase intention is believed to have an objective social reality that can be quantified, which is why it falls under the quantitative research approach [83]. The study examines some of the actionable and quantifiable components like source credibility, e WOM, perceived risk, trust in the online community, and purchase intention, alongside generalizable relationships that can be formed between these factors. As an example, the research takes a deductive approach where a theory or scholarly work regarding social commerce and information credibility is used to formulate working hypotheses, which are then validated by collecting

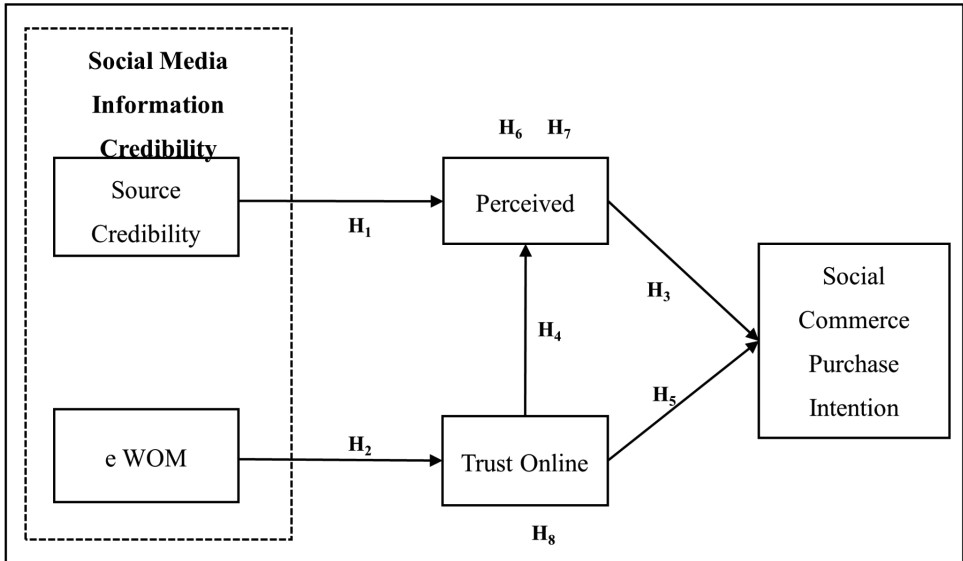

**Fig 2. Conceptual framework.** Source: Authors' compilation.

quantitative data through surveys and statistically analysing the data. It can be said that the combination of these two approaches, positivism and deductive reasoning, provides a comprehensive guide for the research that is socially relevant, empirically tested, and rigorous for validation of findings in relation to social commerce in Sri Lanka [84]. The research approach plays a significant role in a research study at different stages of the investigation. Deductive reasoning was utilised in this study since it defines the steps taken throughout the research process [85].

## Sample and procedure

The target population consisted of Facebook users in Sri Lanka aged 19–34, intending to purchase skincare products through Facebook. This age range was determined based on insights from prior studies, reputable sources and statistical data from reputable sources [77,86,87]. According to Statista [16], Facebook remains the most widely used online platform globally, reinforcing its relevance as the chosen medium for this research. The sample size (n) was calculated to be 384 using Cochran's formula, also based on the highest value from Morgan's table, as referenced in recent studies [7,88,89]. In the case of Sri Lankans who consume skincare products, Facebook has been noted as the most socially and commercially active. In comparison to the other platforms, such as Instagram, TikTok, WeChat, and Facebook provided a more advanced setting for collective engagement, brand interaction, and user recommendations, all vital in building trust and advancing the activities of social commerce. Furthermore, the marketplace capabilities of Facebook, together with its integrated marketing features, position it as the leading platform for the advertising and sale of skincare products. Noted is the fact that differences related to the specific platform may impact consumer behaviour, and other emerging platforms could be researched further. A convenience sampling technique was employed due to the absence of an identifiable sampling frame.

## Questionnaire design

**Measures.** This study's Table 1 captures the relevant variables with their respective dimensions and methods of measurement. Independent variables include source credibility (measured by expertise and trustworthiness) and e WOM credibility (measured by recommendations, reviews, and comments). Social commerce purchase intention (as measured

by social presence, trust, and social interaction) is the sole dependent variable. Meanwhile, trust in an online community (referred to as competence, benevolence, reciprocity, and reliability respectively) alongside perceived risk (financial, time, product, and after-sale risk) serve as mediators. All constructs are measured using a five-point Likert scale, with questionnaire items adapted from well-established prior studies.

**Data collection.** The questionnaire was meticulously crafted to align with the study's research objectives and conceptual framework, incorporating established models referenced in the literature. Designed in both English and Sinhala *(Sinhala, spoken by approximately 74% of Sri Lanka's population, is the most widely used language in the country)* to ensure clarity and inclusivity, the instrument comprised two sections. The first section captured demographic and contextual details pertinent to the study, focusing on participants' age, online skincare product purchasing behaviour, and familiarity with buying skincare products via Facebook. The second section centred on variables integral to the research. Specifically, the study examined two independent variables and two mediating variables, one dependent variable. The study used a five-point Likert scale ranging from "Strongly agree" [1] to "Strongly disagree" [5] in section two, and the primary data were tested for validity and reliability.

There was a thorough two-phase approach in the questionnaire development. Initially, a pilot survey was administered online to a diverse sample of 60 participants. Feedback from this phase, including insights into response patterns and clarity issues, informed refinements to the questionnaire, ensuring improved precision and comprehensibility. Following the pilot study, the finalised questionnaire was utilised to collect all data required for the research study, ensuring robust and comprehensive coverage of the research variables. Participation in the survey was voluntary, and written consent from all the respondents was taken in the first section of the questionnaire. Minimal data set used for the study is available in S1 Appendix. The final version was tested for validity and reliability to enhance data integrity. Survey responses were gathered from 25/08/2024–20/11/2024. The confidentiality of participants is ensured, and the data is anonymised.

## Ethical consideration

To ensure ethical consent, participants were provided with detailed information about the study's objectives and the questionnaire. All concerns were clarified before obtaining consent for participation. Explicit written consent was obtained from each respondent by ticking the consent statement in the questionnaire. Participation was entirely voluntary, and only those who explicitly consented were provided with access to the questionnaire. This consent process was witnessed by the authors. The study was conducted with the ethical approval of the Sri Lanka Institute of Information Technology (SLIIT) Business School Ethics Review Committee (SLIIT/ERC/SBS/2024/07).

## Data analysis

The data analysis in this research utilised the Structural Equation Model (SEM), which is a way of examining and analysing structural relationships. SEM allow this research to model impacts, construct underlying latent variables, identify errors in existing variables, and statistically test existing theoretical measures and assumptions correctly [42,97]. SEM can foster a large sample size, whereby this research targets a minimum sample size of 384 (n = 384). For the final findings, there will be a bootstrapping process with Smart PLS to generate values to test the significance of the model for the hypothesis tests. Smart PLS 4 was chosen over covariance-based SEM tools such as AMOS or LISREL due to the exploratory nature of this study, the non-normal distribution of the dataset, and the complexity of the research model, which includes multiple mediation paths and latent constructs. Variance-based SEM, such as PLS-SEM, is more suitable for prediction-oriented research and models with a higher level of complexity. Moreover, PLS-SEM is robust to violations of multivariate normality. These choices are supported by Hair and Alamer [98], and Ahmed, Streimikiene [89] emphasised the practical advantages of PLS-SEM in similar research contexts. The internal reliability of the responses was assessed using Cronbach's alpha, and descriptive statistics were computed for the two independent variables, two mediating variables, and one dependent variable.

## Data analysis and results

### Demographic profiles

According to Table 2 results, most participants were aged 23–26 years (44%), followed by 19–22 years (23%). Most respondents were female (62%), and in terms of education, a significant portion (58%) had a degree or equivalent higher qualifications. Regarding income levels, nearly half of the respondents (47%) earned below Rs. 50,000. Facebook purchase frequency revealed that 73% of participants made purchases once a month or less, highlighting infrequent buying behaviour on the platform.

### Measurement model assessment

The Fig 3 displays the relationships between constructs along with each indicator. The diagram provides a visual overview of how the observed variables represent their respective latent constructs, supporting the measurement model's reliability and validity.

Cronbach's alpha coefficient, Composite Reliability coefficient, and Average Variance Extracted (AVE) are used to measure the reliability and validity of the model, and the obtained values are shown in Table 3.

The internal consistency of the model variables was assessed using Cronbach's Alpha, with values from 0.751 to 0.912, above the recommended thresholds of 0.7 [89,99]. And Composite Reliability were assessed further, confirming construct reliability with values from 0.836 to 0.945. The validity of the model variables was assessed using AVE with above the recommended thresholds of 0.5 [100,101]. The factor loadings of most indicators were above the

**Table 2. Demographic results.**

| Characteristics | Classification | n | % |
|---|---|---|---|
| Age | 19 - 22 Years | 90 | 23% |
| | 23 - 26 Years | 168 | 44% |
| | 27 - 30 Years | 55 | 14% |
| | 31 - 34 Years | 71 | 18% |
| Gender | Male | 145 | 38% |
| | Female | 238 | 62% |
| | Prefer not to say | 1 | 0% |
| Education Level | Ordinary level | 9 | 2% |
| | Advanced level | 61 | 16% |
| | Diploma level | 91 | 24% |
| | Degree or equivalent higher | 221 | 58% |
| | Prefer not to say | 2 | 1% |
| Income Level | Below Rs. 50,000 | 181 | 47% |
| | Rs.50,000–100,000 | 85 | 22% |
| | Rs.100,000–150,000 | 57 | 15% |
| | Rs.150,000–200,000 | 38 | 10% |
| | Above Rs.200,000 | 23 | 6% |
| Facebook Purchase Frequency | Once a month or less | 282 | 73% |
| | 2-3 times a month | 81 | 21% |
| | Once a week | 10 | 3% |
| | Several times a week | 11 | 3% |

Source: Authors' compilation.

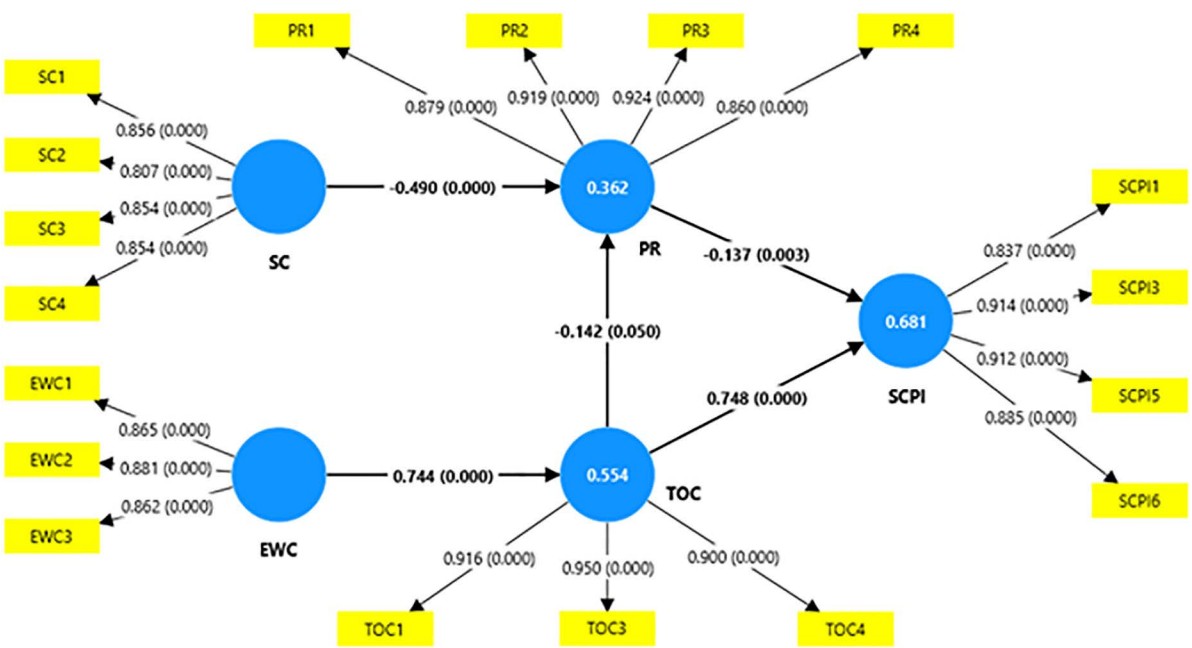

**Fig 3. Graphical output of the SEM measurement model analysis.** Source: Authors' illustration, extracted from Smart PLS.

recommended threshold of 0.7, however, some factor loadings are slightly below but still acceptable [98]. These findings confirm the robustness and quality of the measurement scales used in the study [101,102].

Discriminant validity was assessed using cross-loading, Heterotrait-Monotrait (HTMT) ratio, and Fornell-Larcker criterion, and the obtained values are shown in Tables 4–6.

In Table 4, cross-loadings of each indicator are shown which have the highest loading on its associated construct compared to other constructs. This indicates that all items are strongly related to their respective constructs, confirming discriminant validity based on the cross-loading criterion [98].

According to Fornell and Larcker [101], discriminant validity was further utilised by the Fornell-Larcker criterion in Table 5. Based on this approach, every construct's square root of the average variance extracted (AVE) should be more than its correlation with any other construct. The study confirmed that every construction satisfied this standard. Additionally, the square root of the AVE values was EWC at 0.855, PR at 0.751, SC at 0.806, SCPI at 0.887, and TOC at 0.922. These values exceed the correlations between constructions, significantly the relationship between TOC and SCPI (0.816) and between EWC and PR (−0.701) [101]. The Fornell-Larcker criterion was therefore satisfied and provided further evidence of discriminant validity for the measurement model.

Methodological literature offers various approaches to testing discriminant validity [89]. Among these, the Fornell-Larcker Criterion Fornell and Larcker [101] and cross-loading comparisons have been foundational [103]. However, the Heterotrait-Monotrait (HTMT) ratio introduced by Henseler, Ringle [104] has emerged as a more reliable measure of discriminant validity. Widely adopted in partial least squares path modelling, Ronkko, Mcintosh [105] and structural equation modelling [106]. HTMT is favoured for its robust performance [104]. This study employs HTMT, maintaining a conservative perspective on assessing discriminant validity, as advocated by Ab Hamid, Sami [107] and (refer to Table 6).

**Structural model assessment**

As presented in Table 7, the R values for constructs, such as PR, SCPI and TOC, indicate that there is a high level of ability to explain the variance within the model [108]. Specifically, this model demonstrates strong $R^2$ values, indicating the

**Table 3. Reliability and validity.**

| Construct | Item Code | Survey Question | Factor Loadings | Cron-bach's Alpha | Composite Reliability (rho_a) | Composite Reliability (rho_c) | Average Variance Extracted (AVE) |
|---|---|---|---|---|---|---|---|
| e WOM Credibility | | | | 0.816 | 0.820 | 0.891 | 0.731 |
| | EWC1 | Before I buy a new skincare product through a Facebook page ask their recommendations from my friends on online communities. | 0.852 | | | | |
| | EWC2 | I am willing to recommend a new product that is worth buying for my friends on online communities through commenting and reviews. | 0.860 | | | | |
| | EWC3 | I usually review to buy skincare products on Facebook. | 0.852 | | | | |
| Perceived Risk | | | | 0.751 | 0.798 | 0.836 | 0.565 |
| | PR1 | In my experience, I believe that buying skin care products on Facebook can sometimes lead to money scams. | 0.827 | | | | |
| | PR2 | In my experience, Facebook ads for skincare products sometimes look different from the actual products. | 0.865 | | | | |
| | PR3 | In my experience, the quality of the skincare products on Facebook pages does not always match their price worth. | 0.649 | | | | |
| | PR5 | Based on my experience, communicating with the seller on Facebook pages requires a long time. | 0.636 | | | | |
| Source Credibility | | | | 0.819 | 0.839 | 0.881 | 0.650 |
| | SC1 | I always read Facebook post descriptions when I pur-chase skincare products. | 0.683 | | | | |
| | SC2 | The attractiveness of skin care products posted on Face-book influences my purchase decision. | 0.820 | | | | |
| | SC3 | Based on my experience, sellers on Facebook are very knowledgeable when sharing information related to skin-care products. | 0.866 | | | | |
| | SC4 | In my experience, I believe that sellers who post Face-book content about skincare are skilled and experienced. | 0.843 | | | | |
| Social Commerce Purchase Intention | | | | 0.910 | 0.914 | 0.937 | 0.787 |
| | SCPI1 | I am willing to share my experiences and suggestions with my friends on Facebook when they are buying skin-care products. | 0.838 | | | | |
| | SCPI3 | I am very likely to do transactions via social media for purchasing skincare products. | 0.913 | | | | |
| | SCPI5 | I am willing to search on Facebook for more information about skincare products. | 0.912 | | | | |
| | SCPI6 | If there is the best new skincare product that I want to purchase, I would like to purchase it through Facebook. | 0.885 | | | | |
| Trust Online Community | | | | 0.912 | 0.913 | 0.945 | 0.850 |
| | TOC1 | I believe that sellers of skincare products on Facebook as honest. | 0.916 | | | | |
| | TOC3 | I think the promises made by sellers on Facebook about skincare products are reliable. | 0.950 | | | | |
| | TOC4 | I think that a personal connection with the seller on Face-book influences my decision to buy skincare products. | 0.900 | | | | |

Source: Authors' calculation.

**Table 4. Cross loadings.**

| Constructs | EWC | PR | SC | SCPI | TOC |
|---|---|---|---|---|---|
| EWC1 | 0.852 | −0.513 | 0.588 | 0.613 | 0.576 |
| EWC2 | 0.860 | −0.689 | 0.662 | 0.664 | 0.607 |
| EWC3 | 0.852 | −0.590 | 0.615 | 0.598 | 0.522 |
| PR1 | −0.721 | 0.827 | −0.645 | −0.607 | −0.596 |
| PR2 | −0.646 | 0.865 | −0.705 | −0.624 | −0.650 |
| PR3 | −0.318 | 0.649 | −0.430 | −0.410 | −0.421 |
| PR5 | −0.274 | 0.636 | −0.361 | −0.319 | −0.297 |
| SC1 | 0.552 | −0.474 | 0.683 | 0.430 | 0.397 |
| SC2 | 0.534 | −0.572 | 0.820 | 0.605 | 0.604 |
| SC3 | 0.647 | −0.610 | 0.866 | 0.635 | 0.675 |
| SC4 | 0.615 | −0.709 | 0.843 | 0.628 | 0.688 |
| SCPI1 | 0.652 | −0.584 | 0.589 | 0.838 | 0.633 |
| SCPI3 | 0.653 | −0.603 | 0.668 | 0.913 | 0.795 |
| SCPI5 | 0.660 | −0.583 | 0.649 | 0.912 | 0.719 |
| SCPI6 | 0.638 | −0.644 | 0.647 | 0.885 | 0.738 |
| TOC1 | 0.610 | −0.630 | 0.676 | 0.743 | 0.916 |
| TOC3 | 0.609 | −0.651 | 0.704 | 0.790 | 0.950 |
| TOC4 | 0.627 | −0.613 | 0.686 | 0.724 | 0.900 |

Source: Authors' calculation.

**Table 5. Fornell-Larcker criterion.**

| | EWC | PR | SC | SCPI | TOC |
|---|---|---|---|---|---|
| EWC | 0.855 | | | | |
| PR | −0.701 | 0.751 | | | |
| SC | 0.728 | −0.743 | 0.806 | | |
| SCPI | 0.732 | −0.680 | 0.720 | 0.887 | |
| TOC | 0.667 | −0.685 | 0.747 | 0.816 | 0.922 |

Source: Authors' calculation.

**Table 6. Heterotrait-Monotrait (HTMT) ratio.**

| | EWC | PR | SC | SCPI | TOC |
|---|---|---|---|---|---|
| EWC | | | | | |
| PR | 0.823 | | | | |
| SC | 0.891 | 0.890 | | | |
| SCPI | 0.849 | 0.784 | 0.826 | | |
| TOC | 0.771 | 0.784 | 0.850 | 0.893 | |

Source: Authors' calculation.

independent variables effectively elucidate most of the variation in the dependent variable [89,108]. Also, the slightly lower adjusted $R^2$ values consider the complexity of the model to confirm its stability and durability, while mitigating the potential overfitting [109]. $Q^2$ predict values are used to check whether the model has good predictive ability. In this study, the $Q^2$

**Table 7. Coefficient of Determination (R²), Adjusted R², Predictive Relevance (Q²), and RMSE.**

|  | R-square | R-square Adjusted | Predictive power (Q²) | RMSE |
|---|---|---|---|---|
| PR | 0.591 | 0.589 | 0.582 | 0.649 |
| SCPI | 0.694 | 0.692 | 0.545 | 0.680 |
| TOC | 0.445 | 0.444 | 0.439 | 0.753 |

Source: Authors' calculation.

predicted values are all positive and well above zero, indicating that the model has strong predictive relevance for these constructs [89,98]. RMSE values show the average amount of prediction error. The RMSE values are reasonable and within an acceptable range for social science research [110]. Lower RMSE values mean better prediction accuracy, and in this case, the values are acceptable, confirming that the model predictions are dependable.

The effect size (f²) results indicate that EWC to TOC and TOC to SCPI have large effects, while SC to PR shows a small to medium effect [89]. Other relationships demonstrate small effect sizes, shown in Table 8. Overall, the model's key paths exhibit meaningful impacts according to guidelines.

Furthermore, as seen in Table 9 the model fit indices show a generally good fit, with a focus on the Standardised Root Mean Square Residual (SRMR) and Normed Fit Index (NFI) values [104]. The SRMR values are 0.081 for the saturated model and 0.118 for the estimated model, which is slightly above the ideal threshold of 0.08 but still suggests a good fit [104,111,112]. The d_ULS (Unweighted Least Squares discrepancy) and d_G (Geodesic discrepancy) are higher in the estimated model (2.371 and 0.582, respectively), reflecting greater discrepancies [113]. The Chi-square values of 1031.163 (saturated) and 1146.709 (estimated) suggest some misfit, as lower values are preferred. The NFI decreased slightly from 0.808 to 0.787, yet remains above the acceptable threshold of 0.80, indicating that the model fits the data well [89].

**Table 8. Effect size (f²).**

| Path | F- square (F²) | Effect Size Interpretation |
|---|---|---|
| EWC→TOC | 0.802 | Large Effect |
| PR→SCPI | 0.090 | Small Effect |
| SC→PR | 0.297 | Small to Medium Effect |
| TOC→PR | 0.094 | Small Effect |
| TOC→SCPI | 0.754 | Large Effect |

Source: Authors' calculation.

**Table 9. Results of model fit indices for structural equation modelling.**

| Model Fit Index | Saturated Model | Estimated Model |
|---|---|---|
| SRMR | 0.081 | 0.118 |
| d_ULS | 1.116 | 2.371 |
| d_G | 0.468 | 0.582 |
| Chi-square | 1031.163 | 1146.709 |
| NFI | 0.808 | 0.787 |

Source: Authors' calculation.

**Structural model.** Using Smart PLS 4.0, the conceptual model of research is analysed, which is shown in Fig 4. Bootstrapping with 5000 subsamples was employed to test and validate the hypothesis model [78].

For each measurable question item, factor loadings are depicted in the model above. As shown in the analysis, the lowest PLS factor loading value was 0.636 (> 0.700). The factor loadings for the constructs SC, EWC, PR, TOC, and SCPI demonstrate consistent strength, underscoring robust measurement reliability. All constructions exhibit satisfactory internal consistency, thereby validating the measurement model.

Table 10 reveals significant relationships between the constructs in the model. EWC has a strong positive effect on TOC. PR negatively impacts SCPI, indicating that higher risk perception diminishes consumers' purchase intention. SC negatively influences PR, meaning that a credible source lowers perceived risk. TOC has a significant negative relationship with PR, implying that greater trust reduces the perceived risk. Furthermore, TOC strongly and positively influences SCPI, reinforcing its critical role in shaping purchase intentions.

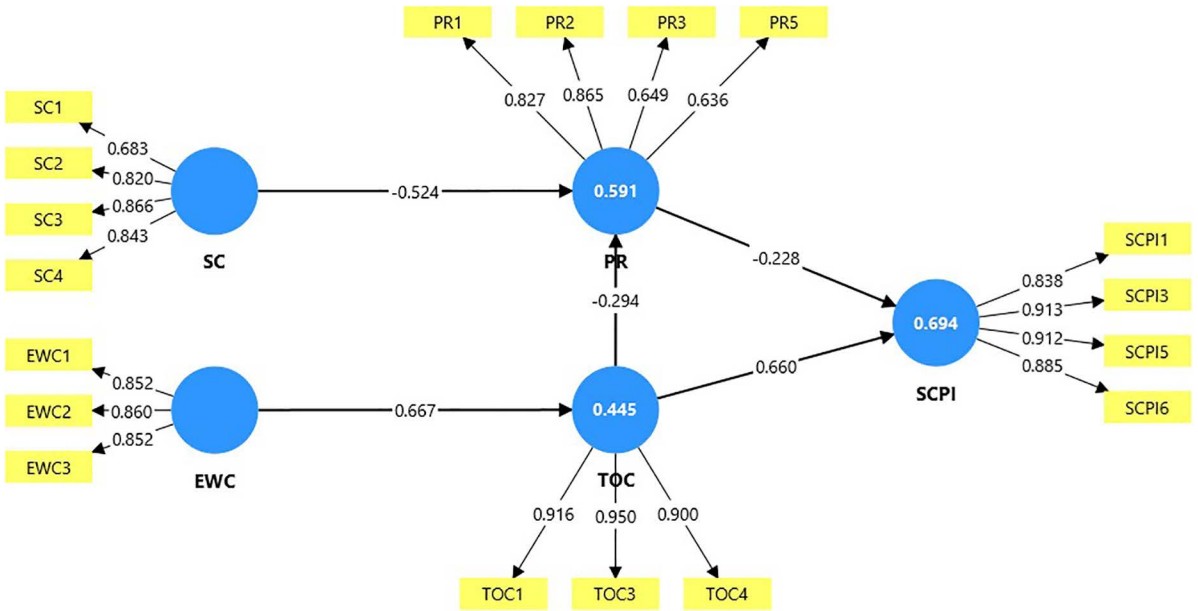

**Fig 4. Graphical output of the structural model analysis.** Source: Authors' illustration, extracted from Smart PLS.

**Table 10. Results of hypotheses analysis.**

| Path | Coefficient | Sample Mean | SD | Confidence Intervals | | t-Statistics | p-Values | Test Results |
|---|---|---|---|---|---|---|---|---|
| | | | | 2.50% | 97.50% | | | |
| EWC→TOC | 0.667*** | 0.669 | 0.032 | 0.603 | 0.73 | 20.661 | 0.000 | Supported |
| PR→SCPI | −0.228*** | −0.234 | 0.070 | −0.373 | −0.101 | 3.283 | 0.001 | Supported |
| SC→PR | −0.524*** | −0.528 | 0.060 | −0.645 | −0.413 | 8.771 | 0.000 | Supported |
| TOC→PR | −0.294*** | −0.293 | 0.070 | −0.423 | −0.148 | 4.183 | 0.000 | Supported |
| TOC→SCPI | 0.727*** | 0.727 | 0.046 | 0.513 | 0.786 | 15.876 | 0.000 | Supported |

Source: Authors' calculation.

Note - *** p-value <0.001 significant at the 1% level

Table 10 results suggest that the relationships between the constructs are robust and statistically significant, with p-values ≤ 0.001 in all cases [105]. The 99% confidence intervals for all path coefficients fall entirely above or below zero, indicating the absence of a zero within the interval range [98,113]. This confirms the statistical significance of all hypothesised relationships between the constructs. The t-statistics for all structural paths exceed the critical value of 1.96, indicating that all relationships in the model are statistically significant at the 5% level. The strong path coefficient ($\beta = 0.727$, p-value = 0.000) between TOC and SCPI underscores the importance of trust in online communities in influencing purchase intention. Similarly, the negative path from SC to PR highlights the significance of source credibility in reducing perceived risk. Overall, the model demonstrates strong predictive power and reliability, confirming that the indicators of each construct make distinct and meaningful contributions without redundancy [104].

**Mediation effect.** The significant indirect and total effects obtained from the bootstrapping procedure in Table 11 highlight with a sample size of 384 (n = 384), using 5,000 subsamples and no sign changes. The use of partial least squares structural equation modelling to capture complex mediation relationships can be effectively demonstrated [114]. The results indicate that source credibility significantly impacts social commerce purchase intention through perceived risk, with a path coefficient. Similarly, e WOM credibility shows a significant indirect effect on purchase intention via trust in the online community and perceived risk. Furthermore, e WOM credibility demonstrates a strong direct effect on purchase intention through trust in the online community alone, with a path coefficient. These findings emphasise the critical mediating roles of trust and perceived risk in shaping purchase intentions in the context of social commerce.

The confidence intervals, which indicate the range in which the true indirect effect is likely to fall with 95% certainty, do not include zero in any of the tested paths, confirming significance. The standard deviation values are relatively low, suggesting consistent estimates across the bootstrapped samples. Furthermore, the t-statistics for all paths exceed the critical value of 1.96, reinforcing the statistical reliability of the mediation effects. Specifically, the paths from SC through PR to SCPI and from EWC through TOC to SCPI are both supported, while the sequential mediation path from e WOM Credibility through TOC and PR to SCPI, although significant, shows a relatively smaller effect size and marginal significance.

## Discussion

Source credibility and e WOM play pivotal roles in shaping consumer perceptions and behaviours within social commerce [6]. In this study, source credibility was found to be instrumental not only in reducing perceived risk but also in fostering reliability and authenticity. This is particularly significant in the skincare industry, where trust in the information source directly influences perceptions of product quality and effectiveness. Aligning with previous findings, credible influencers and representatives who demonstrate expertise, transparency, and consistency are more likely to enhance customer retention and loyalty, further supported by the trust economy [115]. For instance, skincare influencers leveraging scientific evidence or certifications substantially affect consumer decision-making, reducing uncertainties and enhancing trust [116].

Similarly, the study highlights the impact of e WOM credibility in cultivating trust within online communities. User-generated reviews and shared experiences emerge as critical factors that instil consumer confidence, thereby increasing

**Table 11. Results of hypothesis analysis.**

| Path | Coefficient | Sample Mean | SD | Confidence Intervals | | t-Statistics | p-Values | Test Results |
|---|---|---|---|---|---|---|---|---|
| | | | | 2.50% | 97.50% | | | |
| SC→PR→SCPI | 0.120*** | 0.122 | 0.035 | 0.056 | 0.192 | 3.438 | 0.001 | Supported |
| EWC→TOC→PR→ SCPI | 0.045*** | 0.047 | 0.021 | 0.012 | 0.094 | 2.100 | 0.036 | Not Supported |
| EWC→TOC→SCPI | 0.440*** | 0.439 | 0.057 | 0.326 | 0.551 | 7.687 | 0.000 | Supported |

Source: Authors' calculation.

Note - *** p-value <0.001 significant at the 1% level.

purchase intentions [6,97]. This corroborates past research emphasising the role of e WOM in fostering a sense of community trust and social presence, both of which are integral to consumer engagement in social commerce [73,80]. By integrating these dynamics, this study reinforces the importance of credibility and trust as drivers of purchase behaviour, particularly in environments where perceived risk and trust interplay significantly influence decision-making [81].

Source credibility and e WOM play pivotal roles in shaping consumer perceptions and behaviours within social commerce, aligning with the findings of this study. The negative relationship between source credibility and perceived risk, as observed, supports prior research emphasising that credible sources mitigate uncertainties, particularly in contexts like the skincare industry [46]. This aligns with studies suggesting that trust in credible influencers, who showcase expertise and transparency, reduces perceived risk and fosters consumer confidence, thereby enhancing customer retention and loyalty [84,94]. Similarly, the results demonstrate that e WOM credibility positively influences trust in online communities, corroborating existing literature, which highlights the importance of user-generated content in fostering social presence and collective trust within online environments.

Moreover, the finding that perceived risk negatively impacts social commerce purchase intention aligns with prior studies indicating that consumer apprehensions, such as financial or product-related risks, act as barriers to purchase behaviours [79,115]. In addition, the inverse relationship between trust in online communities and perceived risk supports earlier work showing that trust serves as a critical mechanism to alleviate uncertainties in social commerce [51]. Finally, the results confirm that trust in online communities positively influences social commerce purchase intention, echoing previous findings, underscoring trust as a central driver of engagement and transactional behaviours in online platforms [17]. Especially, the findings of this study support the mediation roles of perceived risk and trust in the online community in influencing social commerce purchase intention. The data show that perceived risk mediates the association between source credibility and social commerce purchase intention, implying that increasing source credibility might effectively reduce perceived risk, hence encouraging purchase behaviour [42]. Similarly, trust in the online community mediates the effect of e WOM credibility on perceived risk and social commerce purchase intention, emphasising the significance of trust in determining customers' perceptions of risk and subsequent purchasing decisions. Furthermore, the sequential mediation effect, in which trust first affects perceived risk and then influences social commerce purchase intention, highlights the intricate pathways by which credibility determinants drive consumer behaviour in social commerce environments. Past studies examined these mediation variables' impacts separately; however, this study investigated those mediating impacts together in one investigation with novel insights to the research area [24,42,79,80].

Pivotal, Sri Lanka's cultural considerations need to be thought of while interpreting these results. The Sri Lankan culture is collectivist in nature, which means Sri Lankans strongly identify with their community and are influenced by peers' opinions, which increases the effect of e WOM and source credibility on consumption choices. Such consumers do place trust in community recommendations. Furthermore, shoppers in collectivist societies tend to be more sensitive to perceived authenticity and expertise than shoppers from individualistic societies like the United States and some European countries [29]. Authors from Western markets seem to focus more on the trust a brand receives from the individual, while Sri Lankan consumers, like other Asian markets such as China and India, tend to give more importance to trust received from peers and social networks [6]. Such a cultural focus enhances the impact of trust that online communities have, increasing the mediation effects discussed in this study.

Cross-regional studies have shown these differences are quite striking. Studies from Europe and North America seem to indicate that consumers pay the most attention to intricate details and official documentation when assessing credibility [38,117]. In contrast, trust among Asian consumers, including Sri Lankans, is mostly shaped by social influences and personal relationships [118]. For instance, a study conducted on peer-to-peer commerce in Korea noted that peers' administrator reviews were much more trusted than any corporate statements, which supports the current research. On the other hand, European consumers of skincare were found to be more strongly driven in their trust by regulatory guarantees and brand value, demonstrating how trust systems are potentially culturally very diverse [29].

From the point of view of an industry, these findings provide strategic steps for skincare businesses in Sri Lanka and comparable markets. Businesses can adopt newer technologies to build trust with consumers and manage risk perception. For example, product reviews and commentary from influencers could be authenticated through blockchain technology, thereby increasing trust and reducing doubts about endorsement of transparency [54]. These systems could assure consumers that the feedback provided would not be misleading and would be accurate. The same can be said for AI recommendation systems that suggest products based on validated data and purchase history. They personalise the shopping experience while enhancing credibility. The use of such technologies would improve consumer trust as well as sustain strong brand loyalty from consumers in crowded markets, fostering healthy competition within social commerce markets [10,119,120].

Nonetheless, it is worth noting that some previous studies do not corroborate the outcomes of this research. Some experts believe that the concept of source credibility does not entail a lower perceived risk, nor even a lower risk perception, if there are suspicions that the endorser of an influencer marketing campaign is driven by profit, or if there is an overabundance of marketing promotions. In the same manner, the studies that are more on the pessimistic side of the spectrum assert that e WOM is positively correlated with trust, an emphasis that is needed because there is an overabundance of user-generated reviews that can, at times, lead to a state of confusion, scepticism, or even an overage of information, ultimately decreasing the trust of the consumer. In some instances, due to a more formal reliance on certificates and sophisticated control mechanisms, the so-called peer-generated credibility may be of a lower order. The differences outlined above affirm that the effects of source credibility and e WOM credibility are not universal; they are influenced by attitude, the general level of consumer education, and the prevailing industry standards. As a result, the new findings outlined above ought to be treated more delicately; particularly, they should be treated with the understanding that the factors of source credibility will not always mitigate risk and increase trust in all situations concerning social commerce.

Specifically, research applies the Elaboration Likelihood Model (ELM) within the context of social commerce, illustrating how source credibility alongside e WOM credibility impacts purchase intention via trust and perceived risk. The findings confirm that both aspects of credibility affect risk and trust in a community's e WOM environment, confirming both peripheral and central processing routes of ELM. Additionally, the study reveals new mechanisms of trust and perceived risk while adding to the literature on how consumers process social information and develop purchase intentions. These findings enhance existing literature by outlining the impact of various credibility factors through social commerce behaviour while integrating trust and risk as pivotal factors within ELM-guided consumer decision-making.

Yet, failure to acknowledge shortcomings can lead to misguided conclusions. The research only analyses Facebook users from Sri Lanka, which constrains the applicability of results across other platforms and nations. Subsequent studies can investigate comparisons between platforms and apply longitudinal approaches to studying the changing dynamics of trust and credibility over time.

In conclusion, the findings affirm the importance of information credibility in social commerce actions while highlighting the need for marketing adaptation at a cultural level. Companies that focus on trust cultivation strategies, exploit digital advancements such as blockchain and AI, and customise their marketing strategies according to the culture, stand to benefit more in the rapidly shifting world of digital commerce.

## Implication

### Theoretically implication

Firstly, this research shows both an assessment and exploration of the driving factors of purchase intention in social commerce (source credibility, e WOM credibility, perceived risk, trust online community). Drawing from relevant literature, extensive research on the themes of social commerce purchase intention has been conducted. Existing studies mainly focused on several credibility factors separately but there was a lack of studies that provide a unique framework by using the main utmost credibility factors (e WOM, source) and mediating factors (perceived risk, trust

online community), which are significantly important in determining the impact of social commerce purchase intention. Therefore, this study predominantly empirically highlights indicators that all the factors influence consumer purchase intention. This research builds upon theories related to consumer behaviour concerning social commerce. Using the Elaboration Likelihood Model (ELM), the study's results suggest that different credibility factors work through different routes of persuasion. For instance, Source Credibility and e WOM Credibility operate as peripheral cues and require little expenditure of mental energy to affect purchase intention. In contrast, trust in online communities and perceived risk utilise the central route, which demands greater mental effort to process information. Such findings help polish the application of ELM in online contexts by illustrating that both the rational and emotional pathways influence decision-making regarding social commerce. Earlier research on social commerce has centred on trust, source credibility, or e WOM individually, paying little attention to cognitive frameworks like the ELM. This study adds value by using ELM to describe how both source and message credibility impact purchase intention in social commerce via trust and perceived risk. Incorporating sequential mediation (trust to perceived risk) presents a unique cognitive route that has not been highlighted in the literature on social commerce, thereby broadening theoretical perspectives of online influence among peers [24]. This also helps refine Source Credibility Theory, showing that trustworthiness of the source, together with the message credibility, plays a critical role in shaping consumer trust, which affects intention to purchase. Accordingly, this study strengthens existing theoretical frameworks by broadening their use to include Facebook shopping related to skincare products in Sri Lanka.

## Managerial implication

According to the study's findings, skincare businesses in Sri Lanka that use Facebook can enhance social commerce purchase intention by implementing targeted strategies. Since source credibility has a negative impact on perceived risk, it is essential to collaborate with trustworthy and knowledgeable influencers or content producers to allay consumer concerns. This can involve collaborating with dermatologists, skincare specialists, or regional influencers in addition to displaying authentic reviews and product usage testimonies. Furthermore, businesses should encourage satisfied customers to post reviews, ratings, and testimonials since e WOM fosters trust in the online community. Increasing customer confidence is facilitated by promoting user-generated content and vibrant, encouraging online communities.

The results have implications for users working with social commerce. Marketers and businesses need to work towards improving the perceived trustworthiness of both the brand endorsers and the e WOM content that is disseminated in the online communities. Focus on fostering constructive behaviours like the active encouragement of customer reviews, open exchanges, and peer engagements will serve to enhance confidence and lower risks among consumers, which will lead to an increase in purchase intentions. Given the importance of Facebook in driving purchases of skincare products, brands need to advertise and market on Facebook groups and community pages where there is targeted advertising to promote these products. Moreover, practitioners need to understand other behaviours associated with other platforms because what works for Facebook does not work with Instagram and TikTok. This looks to not only improve strategic decisions that marketers make but also highlights the need with which to approach social commerce within its ever-changing marketplace.

Marketers and social commerce platform managers, especially those operating in emerging markets like Sri Lanka, will find this study informative. It emphasises how trust and perceived risk, two vital components to improve purchase intentions, are influenced by source credibility and e WOM trustworthiness. Managers are advised to cultivate credible sources that encourage user-generated content and trust within online communities. Trust can be strengthened alongside social commerce integration through reliable reviews and transparent communication that effectively diminishes perceived risk [32,33]. Apart from enhancing consumer engagement and conversion rates, these strategies provide social commerce businesses with a competitive edge, distinct from innovation.

## Conclusion

This study was conducted to assess the impact of social media information credibility factors on social commerce purchase intention, with a special focus on Sri Lanka. The research aimed to explore four key variables: source credibility, e WOM credibility, perceived risk, and trust in the online community and their influence on consumers' purchase intentions within the social commerce context. Drawing insights from disciplines such as marketing and psychology, the study confirmed that credibility factors play a significant role in shaping consumer behaviour in social commerce.

By addressing an existing empirical gap and applying theories of information credibility and social commerce purchase intention, this research specifically examined skincare purchasers in Sri Lanka. The findings contribute to the commercial landscape of the country by offering insights that can guide future research in consumer behaviour. For marketers, the study provides valuable implications for developing effective advertising strategies and enhancing product credibility on social media platforms. Ultimately, this research advances the understanding of how social media information credibility influences purchase intention in social commerce settings, with all examined factors demonstrating a notable impact.

This case study captures the issues and prospects of the Sri Lankan skincare industry, noting that brands can improve social media purchase intention by bolstering credibility through influencer marketing, reviews from verified users, and clear disclosure. Facebook was selected as the regional leader for social commerce in Sri Lanka, although other platforms should be compared in further research. This study applies these theories of credibility with ELM in an emerging market context for the first time to show the effect of social media review credibility on purchasing intention and contributes theoretically. These findings are useful for advertisers in strategy development, trust, and risk mitigation. Lastly, the research claims attention for further cultural investigation, international studies, and more diverse sampling for better generalisation.

## Limitations and future research

This research has a few potential gaps that could affect its findings and make them less generalizable. First, the use of convenience sampling, while useful, is associated with selection bias; therefore, the sample's representativeness is hampered. Selecting respondents from Sri Lanka might not guarantee the representativeness of social commerce users globally, hence their relevance in broader contexts. Subsequent research might consider enhancing sample representativeness with probabilistic sampling techniques. Furthermore, the emphasis on Facebook as the only platform for research limits the relevance of the results to other social media sites, while the cited medium is the most accessed social media site for Sri Lankan skincare consumers, social interactions, perceptions of trust, and purchase behaviour likely differ on Instagram, TikTok, or WeChat. As a result, the conclusions might not be true for users of other social media ecosystems. The accuracy and response reliability, bias, and omissions to self-reported metrics are issues addressed in this study. Furthermore, the results are not generalised and are posited to pertain to only Sri Lankan users of Facebook, in addition to the other platforms and cultural environments.

The demographic concentration of the study on persons between the ages of 19 and 29 further reduces the generalizability of the results since it does not reflect the attitudes and behaviours of older users who might have different purchasing behaviour. Additionally, it could be advanced how ELM was integrated within the study, as ELM was cited as a theoretical framework. In this case, source e WOM credibility can be regarded as a peripheral construct that requires low engagement mental effort towards attitude. Conversely, trust in the community and the perceived risk are more likely to align with central processing. Subsequent studies might more clearly define those pathways to shed light on how consumers make decisions.

Furthermore, the choice of elements in the study is limited to source and e WOM credibility, so excluding other possible influences of social commerce purchase intention, such as brand reputation or product quality. The regional and cultural specificity of the research, which concentrates just on Sri Lankan consumers, could also bring prejudices that restrict its relevance to other cultural environments or nations. Finally, considering the fast-changing character of social media, the

results of the study might not be very relevant over time since changes in platform dynamics and user behaviour could affect social commerce interactions and credibility impressions. Future studies could overcome these constraints by broadening the sampling strategy, including a more varied spectrum of platforms, and considering other factors to produce a more complete knowledge of social commerce purchase intention.

Future research could address these limitations by comparing credibility and purchase intention factors across various social media platforms to determine whether platform characteristics influence the roles of source and e WOM credibility. Longitudinal studies could also provide insights into how credibility and trust dynamics evolve and impact purchase intentions. Additional mediators, such as consumer involvement, or moderators, like cultural influences, could be explored to gain a more nuanced understanding of credibility's impact in diverse contexts. Cross-industry analysis would be valuable in assessing how these credibility factors function in different product categories, while experimental designs could offer a controlled examination of causality between credibility and consumer behaviour. Qualitative research, such as interviews or focus groups, could also yield in-depth insights into consumer motivations regarding credibility and trust in social commerce.

In summary, this study highlights the critical roles of credibility and trust in shaping social commerce purchase intentions, while offering a foundation for further research that could broaden and deepen the understanding of consumer behaviour in social media-driven commerce. Years test the impact on newer generations like the emerging Gen Alpha. Furthermore, the current study accommodates future studies with a ground to further assess and evaluate brands and products in Sri Lanka that fall under the nostalgic experiences that function as stimulating agents. Therefore, it is advised that future research undertake practical examples in Sri Lanka and assess consumers' purchase intentions, thereby providing a foundation for businesses and a premise for future manufacturing and marketing practices.

## Supporting information

**S1 Appendix. Minimal data set.**
(XLSX)

## Author contributions

**Conceptualization:** Prarthana Ranjith, Sumudu Nisansala, Nimesha Jayasingha, Kavindya Weerasekara, Krishantha Wisenthige, Nirmani Dayapathirana.

**Data curation:** Prarthana Ranjith, Sumudu Nisansala.

**Formal analysis:** Prarthana Ranjith, Sumudu Nisansala, Nimesha Jayasingha, Kavindya Weerasekara.

**Investigation:** Sumudu Nisansala, Nimesha Jayasingha.

**Methodology:** Prarthana Ranjith, Sumudu Nisansala, Krishantha Wisenthige.

**Project administration:** Krishantha Wisenthige, Nirmani Dayapathirana.

**Software:** Prarthana Ranjith.

**Supervision:** Krishantha Wisenthige, Nirmani Dayapathirana.

**Validation:** Nimesha Jayasingha, Krishantha Wisenthige.

**Visualization:** Prarthana Ranjith, Nimesha Jayasingha, Kavindya Weerasekara.

**Writing – original draft:** Prarthana Ranjith, Sumudu Nisansala, Nimesha Jayasingha, Kavindya Weerasekara.

**Writing – review & editing:** Prarthana Ranjith, Sumudu Nisansala, Kavindya Weerasekara, Krishantha Wisenthige, Nirmani Dayapathirana.

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
