## [Decision Letter · Decision Letter 0]

15 Apr 2025

PONE-D-25-08431Does Social Media Information Credibility Influence Social Commerce Purchase Intention of Skincare Products ? Evidence from FacebookPLOS ONE

Dear Dr. Wisenthige,

Thank you for submitting your manuscript to PLOS ONE. After careful consideration, we feel that it has merit but does not fully meet PLOS ONE’s publication criteria as it currently stands. Therefore, we invite you to submit a revised version of the manuscript that addresses the points raised during the review process.

We look forward to receiving your revised manuscript.

Kind regards,

Anu Sayal, Ph.D.

Academic Editor

PLOS ONE

Journal Requirements:

2. In the online submission form, you indicated that data used in this study available on request without any restriction 

Reviewers' comments:

Reviewer's Responses to Questions

**Comments to the Author**

1. Is the manuscript technically sound, and do the data support the conclusions?

Reviewer #1: Partly

Reviewer #2: Yes

Reviewer #3: Partly

Reviewer #4: Partly

2. Has the statistical analysis been performed appropriately and rigorously? 

Reviewer #1: No

Reviewer #2: Yes

Reviewer #3: No

Reviewer #4: Yes

3. Have the authors made all data underlying the findings in their manuscript fully available?

Reviewer #1: Yes

Reviewer #2: Yes

Reviewer #3: No

Reviewer #4: No

4. Is the manuscript presented in an intelligible fashion and written in standard English?

Reviewer #1: Yes

Reviewer #2: Yes

Reviewer #3: Yes

Reviewer #4: Yes

5. Review Comments to the Author

Reviewer #1: The topic is interesting and, the research paper has wider theoretical and practical applications. The authors have put their best efforts to incorporate the suggested changes. However, I have the following reservations and suggestions for the sake of improvement of the undertaken study:

• The logical sequence of the abstract should be as 1) objectives, 2) methodology, 3) Findings, 4) conclusion and 5) implications. Thus, the authors should also rewrite the abstract in this sequence.

• The authors did not establish the motivation, significance, and novelty of the undertaken study. The authors are suggested to improve this important factor in the "Introduction" section. The structure of the paper is also missing.

• The literature should be presented in an audit form, and it should be linked with the objectives and research question. Moreover, the current and relevant citations should be added in introduction, review of literature and discussions sections. Moreover, the study has adopted a deductive approach, thus, the relevant theory should be integrated with the operational variables. The authors can have the useful insights by studying and citing the following papers:

“Enhancing Competitiveness of E-commerce and the Online Retail Industry via Social Media: Evidence from an AI-Integrated Routine Model”, Journal of Competitiveness, 2024, Vol. 16 (4), 44–59. https://doi.org/10.7441/joc.2024.04.03

“Social Media Marketing of Luxury Brands on Brand Equity, Customer equity and Customer Purchase Intention”, Amfiteatru Economic, 2023, Vol. 25 (62), 265–282. https://doi.org/10.24818/EA/2023/62/265

“Effectiveness of online digital media advertising as a strategic tool for building brand sustainability: Evidence from FMCGs and Services sectors of Pakistan”, Sustainability, 2019, Vol. 11(12), 3436. http://dxdoi.org/10.3390/su11123436

• The methodology should contain the research design, measures, data collection method, and data analysis techniques in separate headings along with the existing sub-headings.

• The PLS-SEM analysis was performed; however, the researchers should have validated the measurement model (outer model) using the reliability and validity analyses using factor loadings, Cronbach’s alpha, composite reliability, average variance extracted for reliability and convergent validity analyses. Similarly, for the discriminant validity the researchers should have used HTMT correlation matrix, Fornell-Larcker criterion, and cross loading analyses. For the validation of structural model (inner model), the coefficient of variation (R2), effect size (f2), predictive power (Q2), Model fitness indices for measurement and structural model besides the coefficient path analysis should be used. The Tables and figures of measurement model and structural model should also be presented. The detailed insights of PLS-SEM modeling could be taken by studying and citing the following paper:

A Comparative Analysis of Multivariate Approaches for Data Analysis in Management Sciences,” E a M: Ekonomie a Management, Vol. 27 (1), 2024, 192–210. https://doi.org/10.15240/tul/001/2024-5-001

• Please follow the sequence as: the discussions section should be added after the results. Followed by theoretical and practical implications, and then limitations and potential areas of future studies.

• The conclusion needs further elaboration and adds at the end of the paper; the conclusion is always one step ahead of the findings.

• The grammatical and spelling mistakes should be corrected.

Reviewer #2: Dear authors,

This study explores the impact of social media information credibility on social commerce purchase intention in the skincare industry, with a focus on Facebook as a primary platform. The study provides valuable insights into the roles of trust, electronic word-of-mouth (eWOM), and perceived risk in shaping consumer purchase behavior. The use of structural equation modeling (SEM) is a strong methodological approach, and the sample size (n = 384) appears sufficient for meaningful analysis.

However, the manuscript requires major revisions before it can be considered for publication. The key areas that need improvement include conceptual clarity, justification of variables, methodological transparency, and discussion depth. Additionally, the generalizability of findings should be addressed, along with better integration of existing literature to strengthen the study’s theoretical foundation.

Major Concerns and Required Revisions

1. Conceptual and Theoretical Clarity

• The study uses source credibility, eWOM, trust in online communities, and perceived risk as predictors of social commerce purchase intention, but it lacks a clear justification of the selection of these specific variables.

Recommendation: Provide a stronger theoretical rationale for why these specific factors were chosen and how they relate to existing models in consumer trust and e-commerce research.

• The Elaboration Likelihood Model (ELM) is briefly mentioned, but the integration of this theory into the study is unclear.

Recommendation: Clearly explain how the ELM framework guides the study and how different credibility factors align with the central vs. peripheral route of persuasion.

2. Methodological Transparency and Justification

• The manuscript states that Facebook was selected as the primary social commerce platform but does not provide a strong justification for excluding other platforms (e.g., Instagram, TikTok, WeChat, etc.), which also play significant roles in the skincare industry.

Recommendation: Explain why Facebook was chosen over other platforms and discuss potential platform-specific biases in consumer behavior.

• The sampling method is described as convenience sampling, which limits the generalizability of findings.

Recommendation: Acknowledge the limitations of convenience sampling more explicitly and discuss how this affects the applicability of the results to broader populations.

• The study examines Sri Lankan consumers, but it does not discuss cultural differences that might affect trust in online reviews and information credibility.

Recommendation: Include a discussion on cultural factors that may influence social commerce behaviors, particularly in comparison to global markets.

3. Data Analysis and Reporting Issues

• While structural equation modeling (SEM) is used, the results section lacks sufficient detail on model validation, fit indices, and robustness checks.

Recommendation:

o Provide model fit indices (e.g., CFI, RMSEA, SRMR) to confirm the suitability of the proposed model.

o Discuss any modification indices or respecifications made to improve model fit.

o Justify why Smart PLS was chosen over other SEM techniques (e.g., AMOS, LISREL).

• The mediation effects of trust and perceived risk are mentioned, but specific indirect effect values and confidence intervals are missing.

Recommendation: Include bootstrap confidence intervals for mediation effects to ensure statistical rigor.

4. Discussion and Practical Implications

• The discussion largely reiterates the results without providing critical analysis or industry implications.

Recommendation: Expand on how businesses can leverage these findings. For example, how can skincare brands improve consumer trust using blockchain verification or AI-driven recommendation systems?

• The manuscript does not compare findings to previous studies in social commerce or skincare marketing.

Recommendation: Provide a deeper comparative discussion with similar studies in Asia, Europe, and North America to highlight differences and similarities.

Reviewer #3: The research is interesting, however, there are a few concerns that need to be addressed. So I suggest to give the author a chance to submit after doing additional research to improve the contributions of the research work. Authors may also consider the following comments for the revision work:

1. For the literature review, authors should refer to more recent research. The current references are not up to date. Authors need to refer to more ISI/Scopus research work instead of the online/conference resources.

2. Authors need to identify the research gap and highlight the contribution(s) of the research work in the manuscript to shows the novelty of the research. Please highlight the motivation of the research in the first and end of the manuscript.

3. More scientific reasoning should be added in the experimental results' explanations. Please provide the critical review for the results obtained.

4. The format of the manuscript needs to be improved. Some sections need to be combined and restructured.

5. The main motivation of the research needs to be mentioned in the manuscript. Not only to describe the existing research work, but to provide a critical review, and a final concluding remark.

6. Please provide justifications for all selected parameters.

7. Please re-check the format of the references and citations.

Reviewer #4: I have carefully reviewed this manuscript and below is my decision.

-I would suggest adding to the literature and referencing it within the introduction and discussion as well. There are studies that have examined e-commerce.

1) https://doi.org/10.1371/journal.pone.0288835

2) https://doi.org/10.14569/IJACSA.2021.0120113

3) https://doi.org/10.14569/ijacsa.2021.0120305

4) https://doi.org/10.5281/zenodo.8429022

5) https://doi.org/10.1177/21582440241287630

- There are many similar studies on this subject. Similar studies are below.

Kumar, A., & Sharma, N. K. (2020). Impact of social media on consumer purchase intention: A developing country perspective. In Handbook of Research on the Role of Human Factors in IT Project Management (pp. 260-277). IGI Global Scientific Publishing.

Kanimozhi, N. V., & Sameera, K. C. (2023). A Study on Impact of Social Commerce on Customer Purchase Intention,'. International Journal of Advanced Research in Science Communication and Technology, 621-625.

Duan, X., Chen, C. N., & Shokouhifar, M. (2024). Impacts of Social Media Advertising on Purchase Intention and Customer Loyalty in E-Commerce Systems. ACM Transactions on Asian and Low-Resource Language Information Processing, 23(8), 1-15.

Govender, K. K., & Yavisha, R. (2023). The impact of Social Commerce on the purchase intentions of millennials using Facebook. Innovative Marketing, 19(2), 223.

Abou Ali, A., Abbass, A., & Farid, N. (2020). Factors influencing customers’ purchase intention in social commerce. International Review of Management and Marketing, 10(5), 63.

You should make the difference of this study clearer by including a section comparing it with similar previous studies.

It can be published after corrections are made.

6. PLOS authors have the option to publish the peer review history of their article (what does this mean? ). If published, this will include your full peer review and any attached files.

**Do you want your identity to be public for this peer review?** For information about this choice, including consent withdrawal, please see our Privacy Policy .

Reviewer #1: **Yes: ** Rizwan Raheem Ahmed, PhD.

Reviewer #2: No

Reviewer #3: No

Reviewer #4: No

---

## [Author Response · Author response to Decision Letter 1]

8 May 2025

Dear Academic Editor and Reviewers,

We would like to express our immense appreciation to you and the honoured reviewers for your positive feedback on the significant improvement in the manuscript. We are further grateful for the insightful comments, which have helped us strengthen the work. Regarding the concerns raised, we have carefully addressed all comments and suggestions. Specifically, the reference list has been thoroughly reviewed and corrected to ensure accuracy and completeness.

Please find attached the revised manuscript, revised manuscript with tracked changes, and authors’ response to the comments & suggestions by reviewers that we are submitting.

After incorporating the changes requested, we believe that this manuscript aligns with the standard of PLOS One.

Thank you once again for your time and consideration. We look forward to your positive response.

Yours sincerely,

Krishantha Wisenthige, Ph.D (Corresponding Author)

---

## [Decision Letter · Decision Letter 1]

23 Jun 2025

PONE-D-25-08431R1Does Social Media Information Credibility Influence Social Commerce Purchase Intention of Skincare Products ? Evidence from FacebookPLOS ONE

Dear Dr. Wisenthige, 

Thank you for submitting your manuscript to PLOS ONE. After careful consideration, we feel that it has merit but does not fully meet PLOS ONE’s publication criteria as it currently stands. Therefore, we invite you to submit a revised version of the manuscript that addresses the points raised during the review process.

We look forward to receiving your revised manuscript.

Kind regards,

Anu Sayal, Ph.D.

Academic Editor

PLOS ONE

Reviewers' comments:

Reviewer's Responses to Questions

**Comments to the Author**

1. If the authors have adequately addressed your comments raised in a previous round of review and you feel that this manuscript is now acceptable for publication, you may indicate that here to bypass the “Comments to the Author” section, enter your conflict of interest statement in the “Confidential to Editor” section, and submit your "Accept" recommendation.

Reviewer #1: All comments have been addressed

Reviewer #5: (No Response)

2. Is the manuscript technically sound, and do the data support the conclusions?

Reviewer #1: Yes

Reviewer #5: Partly

3. Has the statistical analysis been performed appropriately and rigorously? 

Reviewer #1: Yes

Reviewer #5: Yes

4. Have the authors made all data underlying the findings in their manuscript fully available?

Reviewer #1: Yes

Reviewer #5: Yes

5. Is the manuscript presented in an intelligible fashion and written in standard English?

Reviewer #1: Yes

Reviewer #5: Yes

6. Review Comments to the Author

Reviewer #1: The authors have incorporated the suggested changes in the abstract, introduction, literature review, methodology, results, discussion, and conclusion. Therefore, the paper could now be accepted for publication.

Reviewer #5: Thanks for providing me the opportunity to review this manuscript. I hope my comments will be helpful in improving this manuscript.

Introduction: The introduction section does not properly reveal the research gap. What is the novelty of this proposed research in terms of theory and managerial contributions? Major work is required to justify this research's novelty and prospective implications.

• What is the current research progress on the topic? For example, what has been explored before, and why are you pursuing this area?

• What are the theoretical gaps? (e.g., scholarly evidence of contradictory findings and underexplored relationships) and why they should be addressed (e.g., why addressing them is necessary, important, useful, and urgent?);

• What are the practical gaps? (e.g., market and news reports of underlying problems and untapped opportunities) and why they should be addressed (e.g., why addressing them is necessary, important, useful, and urgent?);

• What are the expected contributions to theory? (e.g., what new understanding can be derived?); and

• What are the expected contributions to practice? (e.g., what real-world issues can be solved?).

The authors proposed the research about skin care industry. There must be a brief Sri Lankan skin care industry background, particularly in the context of social commerce. Source credibility of what either consumers or firms? Although the authors discussed UGC in one or two sentences. It’s not enough to justify your arguments. The authors must provide the justifications as a lot of research has been done on the proposed constructs in the context of social media.

The authors generally discussed social media users. How many people use Facebook in Sri Lanka? Social media facts and figures must be related to Sri Lanka not in general terms.

Literature review: First paragraph is irrelevant, must be removed. Instead the authors should focus only on the proposed constructs.

For Source Credibility, the authors already explained that “Balouchi, Aziz (34), has demonstrated a significant relationship between source credibility and perceived risk in the context of online purchasing.” If this relationship is already documented in the literature, what is the novelty of this article?

Similarly, eWOM Credibility (see page 11) “communication information technology have significantly boosted e WOM elements, including reviews, comments, and recommendations, and positively impacted consumers' trust, which certainly strengthens their purchase intentions (6).” The authors must find the gap in the literature to justify this publication.

The proposed model shows mediator only Perceived instead of Perceived Risk. I think it needs to adjust the font to highlight the complete name of the variable.

Which theories support this framework and how?

There must be a theory section which supports this framework and how the proposed theories support this framework.

The theory support is weak and quite unsatisfactory in the context of hypothesis development. The hypotheses must be developed in theory contextualization. It requires major rewriting.

Research design too complex and added unnecessary details. For example, positivism philosophy approach and deductive approach. What is the difference between them. Since the model is too simple, I don’t think such philosophy should be used. It’s a simple quantitative research study.

What was the sampling approach?

For data collection, which social media platforms were used? How did you identify the targeted population?

The authors mainly collected data from users (282) who used Facebook Once a month or less. This creates a serious doubt about the credibility of the findings. Since Facebook is a social media platform where users use it for fun while online purchasing is too complex and users frequently use online platforms to ensure accurate online shopping. Proper justification required.

Table 3 should include the items used for data collection.

The discussion part is weak. It is unclear which results contribute to which theoretical stream and how you position your study's results.

Theoretical implications can be improved. For example, what has been done before? What are the new contributions and their significance in literature?

The managerial contributions can be more emphasized and fine-tuned, given the study context. I would suggest to use some literature support in this part as well.

The quality of communication is average. You need to proofread the whole article to simplify and use concise language and improve the language expression of the article.

7. PLOS authors have the option to publish the peer review history of their article (what does this mean? ). If published, this will include your full peer review and any attached files.

**Do you want your identity to be public for this peer review?** For information about this choice, including consent withdrawal, please see our Privacy Policy .

Reviewer #1: **Yes: ** Rizwan Raheem Ahmed, Ph.D.

Reviewer #5: No

---

## [Author Response · Author response to Decision Letter 2]

9 Jul 2025

Please find the response of the reviewer document provided as a separate file.

---

## [Decision Letter · Decision Letter 2]

3 Sep 2025

PONE-D-25-08431R2Does Social Media Information Credibility Influence Social Commerce Purchase Intention of Skincare Products ? Evidence from FacebookPLOS ONE

Dear Dr. Wisenthige,

Thank you for submitting your manuscript to PLOS ONE. After careful consideration, we feel that it has merit but does not fully meet PLOS ONE’s publication criteria as it currently stands. Therefore, we invite you to submit a revised version of the manuscript that addresses the points raised during the review process.

We look forward to receiving your revised manuscript.

Kind regards,

Md. Rabiul Awal

Academic Editor

PLOS ONE

Journal Requirements:

Additional Editor Comments (if provided):

Reviewer #1:

Reviewer #5:

Reviewers' comments:

Reviewer's Responses to Questions

**Comments to the Author**

1. If the authors have adequately addressed your comments raised in a previous round of review and you feel that this manuscript is now acceptable for publication, you may indicate that here to bypass the “Comments to the Author” section, enter your conflict of interest statement in the “Confidential to Editor” section, and submit your "Accept" recommendation.

Reviewer #1: All comments have been addressed

Reviewer #5: All comments have been addressed

2. Is the manuscript technically sound, and do the data support the conclusions?

Reviewer #1: Yes

Reviewer #5: Partly

3. Has the statistical analysis been performed appropriately and rigorously? 

Reviewer #1: Yes

Reviewer #5: Yes

4. Have the authors made all data underlying the findings in their manuscript fully available?

Reviewer #1: Yes

Reviewer #5: No

5. Is the manuscript presented in an intelligible fashion and written in standard English?

Reviewer #1: Yes

Reviewer #5: No

6. Review Comments to the Author

Reviewer #1: The authors have incorporated the suggested changes in the abstract, introduction, literature review, methodology, results, discussion, and conclusion. Therefore, the paper could now be accepted for publication.

Reviewer #5: I appreciate that the authors have made commendable efforts to address my earlier comments. The manuscript has improved substantially in terms of methodological clarity, data presentation, and integration of relevant literature. While most responses are adequate, some are minimal (“done as suggested”). I encourage the authors to explicitly demonstrate how revisions were implemented, particularly for major methodological and discussion-related comments. Despite improvements, a few points still need tightening before acceptance. I suggest the following revisions to further enhance the manuscript:

1. Terms such as “social commerce purchase intention,” “e-WOM,” and “information credibility” should be used uniformly across abstract, methods, and discussion. Minor inconsistencies remain.

2. Please report exact p-values and effect sizes where applicable, instead of only threshold reporting (e.g., “p < 0.05”). This will enhance statistical transparency.

3. Figure legends should be more self-contained, explicitly including:

sample size (n),

type of statistical test applied,

meaning of significance indicators (*, **, etc.).

4. While expanded, the discussion still leans towards summarizing findings. I recommend:

More critical engagement with contrasting studies, not just supportive literature.

A clearer acknowledgment of limitations (e.g., convenience sampling, generalizability to broader populations, reliance on self-reported data).

5. The manuscript would benefit from a professional language edit to improve grammar, readability, and overall flow.

7. PLOS authors have the option to publish the peer review history of their article (what does this mean? ). If published, this will include your full peer review and any attached files.

**Do you want your identity to be public for this peer review?** For information about this choice, including consent withdrawal, please see our Privacy Policy .

Reviewer #1: No

Reviewer #5: No

---

## [Author Response · Author response to Decision Letter 3]

22 Sep 2025

A detailed response document is provided in response to the reviewers' file.

---

## [Editor Report · Decision Letter 3]

23 Sep 2025

Does Social Media Information Credibility Influence Social Commerce Purchase Intention of Skincare Products ? Evidence from Facebook

PONE-D-25-08431R3

Dear Dr. Wisenthige,

We’re pleased to inform you that your manuscript has been judged scientifically suitable for publication and will be formally accepted for publication once it meets all outstanding technical requirements.

Kind regards,

Md. Rabiul Awal

Academic Editor

PLOS ONE
---

## [Editor Report · Acceptance letter]

PONE-D-25-08431R3

PLOS ONE

Dear Dr. Wisenthige,

I'm pleased to inform you that your manuscript has been deemed suitable for publication in PLOS ONE. Congratulations! Your manuscript is now being handed over to our production team.

Kind regards,

on behalf of

Dr. Md. Rabiul Awal

Academic Editor

PLOS ONE